# Adiponectin Intervention to Regulate Betatrophin Expression, Attenuate Insulin Resistance and Enhance Glucose Metabolism in Mice and Its Response to Exercise

**DOI:** 10.3390/ijms231810630

**Published:** 2022-09-13

**Authors:** Qi Guo, Shicheng Cao, Xiaohong Wang

**Affiliations:** 1Department of Health Promotion, School of Intelligent Medicine, China Medical University (CMU), Shenyang 110000, China; 2Center of 3D Printing and Organ Manufacturing, School of Intelligent Medicine, China Medical University (CMU), Shenyang 110000, China

**Keywords:** diabetes, insulin resistance, adiponectin, betatrophin, exercise

## Abstract

**Aims:** Adiponectin stimulates mitochondrial biogenesis through peroxisome proliferator-activated receptor-coactivator1α (PGC-1α), a major regulator of mitochondrial biogenesis, and its effect on the genesis of insulin resistance is organ-specific. Expressed predominantly in fat and liver tissues, betatrophin is primarily involved in lipid metabolism, and could be a putative therapeutic target in metabolic syndrome and T2D. We hypothesized that the adiponectin pathway may regulate the production and/or secretion of betatrophin in liver. We aimed to determine whether exercise and adiponectin affect betatrophin to improve insulin resistance in mice. **Methods:** To investigate this hypothesis, we used wild-type C57BL/6 mice subjected to a high-fat diet, an exercise regimen, and i.p. injection of recombinant mouse adiponectin (Acrp30), and adiponectin knockout (Adipoq−/−) mice (C57BL/6 background) subjected to i.p. injection of Acrp30. **Results:** In Adipoq–/– mice, betatrophin levels in the plasma and liver were upregulated. In mice, plasma and liver betatrophin levels were significantly upregulated following a high-fat diet. Exercise and i.p. Acrp30 downregulated betatrophin levels and increased adiponectin mRNA and protein expression in the plasma and liver. The trend of change in PGC-1α and betatrophin levels in the liver was consistent. **Conclusions/interpretation:** Exercise reverses pathogenic changes in adiponectin and betatrophin levels in insulin-resistant mice. Exercise increased adiponectin levels and reduced betatrophin levels. Furthermore, exercise reduced betatrophin levels via adiponectin, which modulated the LKB1/AMPK/PGC-1α signaling axis but was not solely dependent on it for exerting its effects.

## 1. Introduction

Diabetes is a worldwide epidemic. According to the International Diabetes Federation (IDF), the worldwide prevalence of diabetes mellitus in 2011 was 366 million with projections that by 2030, the prevalence will reach 552 million with a 7.7% comparative prevalence [1]. Diabetes promises to become an even larger public health issue with a significant social and economic burden with clinical practice and public health policy implications [2].

The main pathogenetic basis of type 2 diabetes mellitus (T2D) is a glucose and lipid metabolism disorder caused by insulin resistance. Therefore, the liver—the center of glucose and lipid metabolism—is the key target organ of T2D research. Controlling blood sugar levels is an effective measure for preventing diabetes or delaying its development. Drug and insulin dependence to control blood glucose levels can be reduced through lifestyle interventions [3]. Recently, betatrophin has garnered considerable research attention as a predictive biomarker for diabetes occurrence, development, and severity. Expressed predominantly in fat and liver tissues, betatrophin is primarily involved in lipid metabolism [4,5]. Although the role of betatrophin in strengthening islet β-cell proliferation [6] is contested, it has a definite role in regulating lipid metabolism [7,8,9]. Therefore, betatrophin could be a putative therapeutic target in metabolic syndrome and T2D. Moreover, it is an independent predictor of T2D [10]. However, the physiological function and molecular targets of betatrophin are largely unknown. A recent report showing that circulating betatrophin levels are negatively correlated with adiponectin concentrations in patients with insulin resistance attracted our attention [11]. Importantly, betatrophin and adiponectin are both biomarkers of diabetes mellitus [12]. Therefore, exploring the molecular machinery involved in adiponectin–betatrophin interactions under insulin resistance could prove extremely valuable both academically and clinically in T2D prevention, diagnosis, and treatment.

Adiponectin plays many metabolic roles; it exerts anti-obesity, anti-inflammatory, and insulin sensitization effects. It negatively correlates with obesity and insulin resistance [13] and improves metabolic effects [14]. Its effects are regulated by the transmembrane receptors AdipoR1 and AdipoR2. Among them, AdipoR2, which can enhance insulin sensitivity, is primarily expressed in the liver. Yamauchi et al. [15] confirmed that AdipoR gene deletion leads to enhanced gluconeogenesis, weakened 5’-AMP-activated protein kinase (AMPK) activity, and reduced glucose uptake. The constitutively active serine/threonine protein kinase liver kinase B (LKB1) is mainly located in the nucleus under normal physiological conditions and plays an important role in adiponectin-induced AMPK activation [16]. Stimulated by adiponectin, LKB1 translocates to the cytoplasm and subsequently induces AMPK phosphorylation and activation. These results could be replicated in vitro using AICAR (Acadesine, 5-aminoimidazole-4-carboxamide-1-β-D-ribofuranoside), an AMPK activator [17].

AMPK is a key molecule that regulates biological energy metabolism and is also at the core of diabetes and other metabolic disease research. It is expressed in various organs involved in metabolism and is activated by various stimuli, including cell stress, exercise, and hormones, which affect cell metabolism. Exercise can increase AMPK activity in close association with PGC-1α in rat skeletal muscle. Thus, activated AMPK can promote PGC-1α transcription and probably its phosphorylation at Thr177 and Ser538. PGC-1α, which cooperates with a variety of transcription factors to regulate adaptive heat production, mitochondrial biosynthesis, and liver gluconeogenesis and affects blood glucose metabolism, may be the main regulator of mitochondrial biogenesis [18]. However, the activity of PGC-1α is not consistent; its effect on the genesis of insulin resistance is organ-specific [19,20,21]; for instance, while it promotes insulin resistance in the liver, it reduces insulin resistance in muscle tissue. These contrasting effects could be due to the enhancing effects of PGC-1α on hepatic gluconeogenesis under insulin resistance [22]. The biology and regulation of PGC-1α activity have not been fully understood. In addition, PGC-1α and betatrophin are associated with energy consumption or heat generation [5,23]. Notably, PGC-1α may indirectly promote betatrophin expression in muscle [24]. However, it is not clear whether adiponectin downregulation under insulin resistance affects the LKB1/AMPK/PGC-1α signaling axis. Moreover, the effect of this axis on betatrophin expression is not known.

Exercise training is effective in treating obesity-induced, insulin-resistant diabetes mellitus. Exercise can improve insulin sensitivity in obese patients by increasing glucose use [25]. The betatrophin levels decrease after exercise; interestingly, this exercise-induced decrease in betatrophin levels occurs only in obese patients [26]. Weight-loss regimens, involving a 6-month calorie restricted diet or diet + activity, resulted in a significant reduction in betatrophin levels [27]. Exercise can reverse the decrease in serum adiponectin level caused by insulin resistance (induced by a high-fat diet (HFD)) and restore it to the normal level [28]. Thus, here we aimed to clarify whether (1) exercise synergizes adiponectin-mediated activation of the LKB1–AMPK–PGC-1α axis and (2) this axis affects betatrophin expression in the liver and improves insulin resistance.

## 2. Results

### 2.1. Effect of Adiponectin KO on Betatrophin Expression in Liver

Protein from mouse liver tissue was extracted and analyzed using western blotting. Plasma Adiponectin expression in the livers of KO mice was significantly lower than that in those of CON mice, whereas betatrophin levels were significantly higher than those in CON mice (Figure 1). This shows that adiponectin affects betatrophin expression, and these two proteins are negatively correlated.

### 2.2. Effects of Exercise on Adiponectin and Betatrophin Expression

Protein expression in tissues obtained from exercising knockout (KO + EXE) and control (CON+EXE) mice was studied using immunoblotting. Exercise increased plasma adiponectin expression of CON mice; lack of adiponectin expression in KO mice was confirmed (Figure 1). Conversely, exercise decreased betatrophin expression in the CON + EXE and KO + EXE groups. When expression across all the groups (CON, CON + EXE, WT, WT + EXE, KO, and KO + EXE) was compared, we found that exercise intervention decreased betatrophin levels in CON and KO mice; both CON + EXE and KO + EXE mice showed significantly lower betatrophin levels than their non-exercising counterparts (CON and KO) (Figure 2). Thus, aerobic exercise increased adiponectin expression and inhibited betatrophin secretion. Moreover, we concluded that aerobic exercise can also regulate betatrophin secretion in an adiponectin-independent manner.

### 2.3. Exercise Affects Betatrophin Expression through the LKB1/AMPK/PGC-1α Pathway

We next quantified betatrophin, Lkb1, and Pgc1α mRNA levels in mouse liver tissue using RT-qPCR. Exercise significantly downregulated betatrophin in the CON + EXE and CON + EXE + Acrp30 groups (Figure 3A) compared to that in the CON group. Compared with the expression in the CON + Acrp30 and CON+EXE groups, exercise, and adiponectin co-intervention (CON + Acrp30 + EXE) did not have a superposition effect on betatrophin expression. In KO mice, exercise consistently lowered betatrophin levels significantly (KO + EXE and KO + Acrp30 + EXE groups vs. KO and WT groups) (Figure 3A). Co-intervention with exercise and adiponectin (KO + Acrp30 + EXE) did not elicit further downregulation of betatrophin mRNA levels in KO mice when compared with those in the groups that received exercise or adiponectin alone (KO + Acrp30 and KO + EXE groups) (Figure 3A). However, whereas the Lkb1 mRNA level increased significantly in CON mice receiving exercise and adiponectin co-intervention (Figure 3A), Pgc1α mRNA content decreased significantly (Figure 3A). Similarly, the Lkb1 mRNA level increased significantly with exercise and adiponectin co-intervention in the KO mice (Figure 3A), and Pgc1α mRNA content decreased significantly (Figure 3A). Lkb1 and Pgc1α were expressed in both the CON WT and KO groups; co-intervention with exercise and adiponectin did not exert a superposition effect on their expression levels.

Protein analysis of CON mouse liver tissues and plasma revealed that intervention with either exercise or adiponectin (CON + EXE or CON + Acrp30, respectively) or co-intervention with exercise and adiponectin (CON + Acrp30 + EXE) upregulated adiponectin and downregulated betatrophin expression (Figure 3B,F). The adiponectin upregulation and betatrophin downregulation in the CON + Acrp30 + EXE group were not significant when compared with those in the CON + Acrp30 and CON + EXE groups (Figure 3B,F). The effect of exercise and adiponectin interventions in the KO mouse groups was like that in the CON group. Compared with those in the KO group, the adiponectin levels in the KO + EXE, KO + Acrp30, and KO + Acrp30 + EXE groups increased significantly, whereas the betatrophin levels decreased significantly; these changes were significant when compared with the expression in the WT group (Figure 3B,F). Adiponectin level in the KO + Acrp30 + EXE group was not significantly different from that in the KO + Acrp30 and KO + EXE groups. Although co-intervention with exercise and adiponectin tended toward a superposition effect on betatrophin levels, they were not significantly affected when compared with those in the KO + Acrp30 and KO + EXE groups (Figure 3B,F). The exercise and Acrp30 interventions seemed to have similar effects. In summary, co-intervention with exercise and adiponectin did not work synergistically.

To elucidate the mechanism underlying the changes in adiponectin and betatrophin levels affected by exercise, LKB1, AMPK, p-AMPK, and PGC-1α levels were estimated using western blotting. CON + Acrp30 + EXE mice showed significantly increased LKB1 and p-AMPK levels (Figure 3C,D), whereas that of PGC-1α, a downstream regulator of p-AMPK, decreased significantly (Figure 3E). The effects of the co-intervention were not significantly different from those of the single interventions. LKB1 and p-AMPK levels in the KO group decreased significantly compared with those in the CON group (Figure 3C,D), whereas the PGC-1α levels increased significantly (Figure 3E). Intervention with either exercise or adiponectin reversed the decrease in LKB1 and p-AMPK levels; KO + EXE and KO + Acrp30 mice had significantly increased levels of LKB1 and p-AMPK (Figure 3C,D); whereas the PGC-1α levels decreased significantly (Figure 3E). Similarly, with the co-intervention, greater recovery from LKB1 and p-AMPK downregulation was seen compared with that in KO + EXE and KO + Acrp30 mice; however, the changes were not significant (Figure 3C,D). Thus, LKB1, p-AMPK, and PGC-1α appear to function downstream of adiponectin; changes in adiponectin levels affected their expression in mouse liver.

### 2.4. Exercise Inhibits Betatrophin Expression and Improves Insulin Sensitivity in Obese Mice

While maintaining the HFD insulin resistance model mice, we recorded their weekly weight change. At 11 weeks, the body weight of HFD mice was significantly higher than that of CON mice. Exercise intervention significantly reduced the body weight of HFD mice; adiponectin intervention did not affect the body weight. However, co-intervention with exercise and adiponectin significantly decreased body weight (Figure 4A). We also assessed fasting blood glucose and insulin concentrations to determine whether HFD induces insulin resistance. Blood glucose and insulin levels in HFD mice were significantly higher than those in CON mice. However, blood glucose concentration in the HFD + Acrp30 group was significantly lower than that in the HFD group, and that in the HFD + Acrp30 + EXE group was significantly lower than that in the HFD + EXE group. Similarly, insulin concentrations were significantly lower in the HFD + EXE and HFD + Acrp30 groups when compared with that in the HFD group and showed a superposition with those for co-intervention (HFD + Acrp30 + EXE). Therefore, it can be preliminarily stated that HFD mice have insulin resistance symptoms, and the insulin resistance model here was successfully established. Here, the liver tissue of insulin-resistant mice was sampled for mRNA and protein analyses. RT-qPCR revealed that whereas Lkb1 mRNA levels in the HFD group decreased significantly, those of Pgc1α and betatrophin increased significantly (Figure 4F). Adiponectin intervention significantly reversed the reduced Lkb1 levels in HFD mice. In addition, it reversed the increases in Pgc1α and betatrophin mRNA levels in HFD mice (Figure 4F). Similarly, exercise reversed the reduced Lkb1 mRNA levels in the HFD group; moreover, Pgc1α and betatrophin levels decreased significantly after exercise intervention (Figure 4F). Co-intervention with exercise and adiponectin significantly increased Adiponectin and Lkb1 and decreased Pgc1α and betatrophin mRNA levels when compared with those in CON and HFD mice. However, when compared with that in the CON + EXE, CON + Acrp30, HFD + EXE, and HFD + Acrp30 groups, the superposition effect was not seen. Western blotting showed that adiponectin, LKB1, and p-AMPK expression in the HFD group was significantly lower than that in the CON group (Figure 4D,G–H), whereas the PGC-1α and betatrophin levels increased significantly (Figure 4I,J). Adiponectin intervention significantly increased adiponectin(plsma), LKB1, and p-AMPK levels in the liver compared with those in the HFD group (Figure 4D,G–J), whereas it decreased those of PGC-1α and betatrophin (Figure 4I,J). Exercise training significantly increased plasma adiponectin, LKB1, and p-AMPK expression when compared with that in the CON group (Figure 4D,G–H) and decreased PGC-1α and betatrophin levels (Figure 4I,J). Co-intervention significantly increased plasma adiponectin, LKB1, and p-AMPK expression compared with that in the HFD group (Figure 4D,G–H). and decreased that of PGC-1α and betatrophin (Figure 4I,J). However, the changes were not significant when compared with the levels in the HFD + EXE or HFD + Acrp30 groups.

To verify the validity of these results, we analyzed the serum protein levels in the HFD and CON groups using ELISA. Plasma betatrophin concentration was significantly higher in the HFD group than in the CON group (Figure 4E). Furthermore, exercise reduced serum betatrophin level in the HFD group. Moreover, adiponectin intervention yielded results like those of aerobic exercise. Notably, with co-intervention, serum betatrophin level was significantly lower than that in the HFD group and showed a significant downward trend compared with that in the HFD + EXE or HFD + Acrp30 group (Figure 4E).

## 3. Discussion

The data presented herein demonstrates that adiponectin regulates the production and/or secretion of betatrophin in the liver and that exercise intervention reduces betatrophin production and improves metabolism in mice. Adiponectin signaling is believed to be an important regulator of liver betatrophin based on the following findings: (1) adiponectin treatment in Adipoq−/− mice reversed the levels of PGC-1α and betatrophin in the liver and increased the liver expression of LKB1 and Ampk; (2) exercise reduced the plasma and liver levels of betatrophin in mice, increased the liver expression of LKB1 and Ampk, and reduced body weight and plasma insulin levels. This study thus supplies evidence to support the hypothesis that adiponectin increases plasma and liver betatrophin levels by increasing LKB1 and Ampk expression and reducing PGC-1α expression in the liver.

Aerobic exercise and adiponectin interventions can reduce betatrophin levels in the livers of control, KO, and insulin-resistant mice. We compared the adiponectin KO and control mice here and found that in the KO mice, adiponectin protein levels were significantly lowered, whereas those of betatrophin, a protein related to lipid metabolism and insulin resistance, were significantly increased. This indicated a negative correlation between adiponectin and betatrophin. Moreover, adiponectin may function as the upstream signal of betatrophin. To further clarify the relationship between adiponectin and betatrophin, we studied the effect of adiponectin intervention. Across the groups, adiponectin intervention decreased betatrophin expression in the liver; thus, adiponectin levels can affect betatrophin secretion. Furthermore, adiponectin intervention had opposing effects on LKB1, p-AMPK, and PGC-1α expression; it significantly increased LKB1 and p-AMPK levels while decreasing PGC-1α levels.

To validate the correlation between aerobic exercise and adiponectin and betatrophin, we studied the effect of aerobic exercise intervention on CON and adiponectin KO mice. Exercise significantly increased adiponectin levels while decreasing those of betatrophin across all the groups studied. Thus, we confirmed that exercise can regulate both adiponectin and betatrophin levels. To understand the role of adiponectin in this effect, we used exercise intervention in adiponectin KO mice; levels of various proteins in the KO and KO + EXE groups were compared. While exercise increased LKB1 and p-AMPK protein expression in the liver, it downregulated PGC-1α expression. This shows that exercise can regulate LKB1, p-AMPK, and PGC-1α adiponectin independently. Since the KO mice lacked adiponectin, p-AMPK secretion decreased and PGC-1α and betatrophin expression increased significantly. Against this backdrop of insufficient adiponectin, we performed aerobic exercise and/or adiponectin intervention experiments. These interventions significantly attenuated the changes induced by adiponectin loss; the p-AMPK level was enhanced, and betatrophin and PGC-1α levels were decreased. These experiments confirmed the conjecture that adiponectin is an upstream signal that triggers changes in LKB1, p-AMPK, and PGC-1α levels. PGC-1α has been implicated as a betatrophin regulator [24]. Therefore, it is safe to assume an internal relationship between adiponectin, LKB1, p-AMPK, PGC-1α, and betatrophin. Thus, exercise and adiponectin can reduce PGC-1α/betatrophin expression in mouse liver by activating LKB1/AMPK.

Plasma Adiponectin expression decreased significantly in the HFD (insulin-resistant) mice in this study, whereas PGC-1α and betatrophin expression was significantly increased in livers. In the comparative experimental analysis of the effect of adiponectin and aerobic exercise interventions on HFD mice, we found that they similarly alleviated the abnormal expression of the proteins. However, we did not find an obvious superposition effect between the two. Notably, the effect on PGC-1α is organ-specific [19,20,21]: PGC-1α promotes insulin resistance in the liver but attenuates it in muscle tissue. This may be related to enhanced hepatic gluconeogenesis due to insulin resistance [22]. The findings of this study can be summarized as follows: exercise and adiponectin interventions can reverse the significant increase in PGC-1α expression in livers of insulin-resistant mice and restore it to the normal level in vivo; changes in betatrophin levels closely reflect those in PGC-1α levels. In addition, changes in serum levels of fasting blood glucose, insulin, adiponectin, and betatrophin are consistent with corresponding changes in liver tissue. Thus, aerobic exercise and adiponectin interventions can alleviate insulin resistance through related signal pathways and play a role in balancing energy metabolism.

## 4. Material and Methods

### 4.1. Mice

The animal allocation and experimental set-up is shown in Figure 5. Twenty 12-week-old, male adiponectin knockout (Adipoq-KO (Adipoq−/−)) mice (C57BL/6 background) and their wild-type littermates were obtained from Shanghai Nanfang Research Center for Model Organisms (Shanghai, China). Pellets were replaced twice weekly, and body weight and food consumption were recorded both daily and weekly.

Normal 5-week-old male C57BL/6 mice (Liaoning Changsheng Biotechnology Co., Ltd.) were randomly assigned to either a standard chow diet (CON, n = 40) or high-sugar HFD (60% kcal from fat, n = 40) feeding. The body weight after 11 weeks of HFD feeding was 20% higher than that of the CON group; thus, an increase in body weight was considered to indicate the successful establishment of the HFD-induced insulin resistance model mice. The CON, HFD, Adipoq-WT littermate (WT, henceforth), and Adipoq-KO (KO, henceforth) groups received intervention in the form of exercise and/or Acrp30 (a recombinant adiponectin) during the last week of exercise training. Animals received either Acrp30 (Novoprotein, China) 28 µg/kg/day intraperitoneally (i.p.) [28] or vehicle (0.9% sodium chloride injection, 0.5 mL/day i.p.) depending on the group they belonged to. The Institutional Animal Care and Use Committee approved all experiments (Laboratory Animal Service Center, China Medical University).

### 4.2. Vector Construction

The Adipoq-KO targeting vector was constructed from a genomic DNA fragment derived from a C57BL/6 genomic bacterial artificial chromosome clone. The targeting vector, the 5ʹ homologous arm (3037 bp), LacZ p (3.6 kb), and 3ʹ homologous arm (3506 bp) were inserted into the XpPNT vector. The linearized Adipoq-KO targeting vector was electroporated into ES embryonic stem cells. The targeted clones were identified using polymerase chain reaction (PCR) analysis. Correctly targeted embryonic stem cell clones were microinjected into blastocysts of C57BL/6 mice to generate chimeras that were then crossed with C57BL/6 genetic mice. The offspring were screened using PCR analysis of genomic DNA using Adipoq wild type (WT)- and Adipoq-KO-specific primers. These were custom generated by Shanghai Nanfang Research Center for Model Organisms (China).

### 4.3. Exercise Training

After a 2-day acclimation to treadmill running, mice were trained to run at 20 m/min for 1 h per day, 5 days per week. After 8 weeks of exercise training and 36–48 h after an exercise session, the mice were subjected to a 12-h fast. Subsequently, tail vein blood glucose level after 12-h fasting was measured using a blood glucose monitoring system (Sinocare, China); plasma and liver tissues were pooled together.

### 4.4. Measurement of Serum Protein Levels

Serum insulin, adiponectin, and betatrophin levels were quantified using an enzyme-linked immunosorbent assay (ELISA) according to the manufacturer’s instructions (mlbio, Shanghai, China). The intra- and inter-assay CVs were 5.1 and 6.5% and 9.28 and 10.2%, respectively.

### 4.5. Western Blotting

Protein lysates from mice were prepared and analyzed using western blotting as previously described [20]. Briefly, the immunoblots were incubated overnight at 4 °C with primary antibodies against phospho (p)-AMPK, AMPK, PGC1α, actin (Abcam, USA), LKB1 (Bioss Antibodies, USA), and betatrophin (Proteintech, USA).

### 4.6. RNA Extraction and Quantitative Reverse Transcription Quantitative (RT-qPCR)

Total RNA from mice was isolated using the TRIzol reagent. For RT, 1 µg total RNA was converted into first-strand complementary (c)DNA in 20 µL reactions using TB GreenTM Premix Ex TaqTM II (TaKaRa, USA). The cDNA was analyzed using RT-qPCR on an SYBR Green QPCR system (Applied Biosystems), and the relative abundance of the genes were estimated compared to that of β-tubulin. Specific primers used for mouse Adipoq, Lkb1, Pgc1α, betatrophin, and β-tubulin are described in Appendix A.

### 4.7. Data Analysis and Statistics

Statistical tests were conducted using GraphPad Prism version 8.0 (GraphPad Software, San Diego, CA, USA). Quantitative data were presented as mean ± SD. Student’s *t*-test was employed to examine the difference between two groups, while ANOVA was used to evaluate the differences among multiple groups. All these tests were two-tailed, and a *p*-value of <0.05 was considered statistically significant.

## 5. Conclusions

Using adiponectin KO mice and exercise and drug interventions, we preliminarily determined that adiponectin is the upstream regulator of betatrophin, and they are negatively correlated in the liver. Secondly, the effect of adiponectin on the LKB1–AMPK–PGC-1α–betatrophin signal cascade was confirmed; adiponectin regulates PGC-1α/betatrophin expression in the liver by affecting LKB1. Notably, exercise and adiponectin intervention can reverse the significant increase in PGC-1α/betatrophin expression in livers of insulin-resistant mice and restore it to normal levels.

Adiponectin and aerobic exercise interventions could regulate the LKB1–AMPK–PGC-1α signaling axis in the livers of all mouse groups studied here. It confirmed that exercise and adiponectin interventions have similar effects on the signal pathways affected in insulin-resistant mice. While aerobic exercise can affect adiponectin expression, its effect on LKB1–AMPK–PGC-1α signaling is not solely dependent on adiponectin. Finally, we confirmed that adiponectin and aerobic exercise regulate betatrophin in the pathogenesis of insulin resistance. This establishes a theoretical and practical foundation for the prevention, control, and cure of type 2 diabetes caused by insulin resistance.

## Figures and Tables

**Figure 1 ijms-23-10630-f001:**
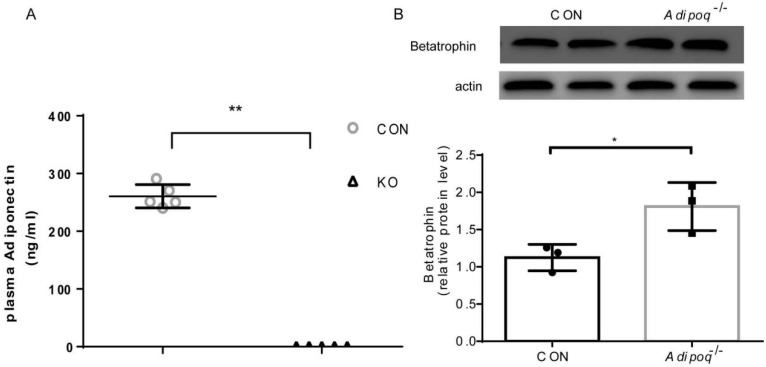
Effect of adiponectin knockout on betatrophin expression. (**A**) Plasma concentrations of adiponectin and betatrophin determined using ELISA at the end of the interventions (n = 6–8 in each group). (**B**) Western blot analysis of adiponectin and betatrophin expression in mouse liver. Data from at least three independent experiments are shown and are expressed as the mean ± standard deviation. Student’s *t* test: * *p* < 0.05, ** *p* < 0.01.

**Figure 2 ijms-23-10630-f002:**
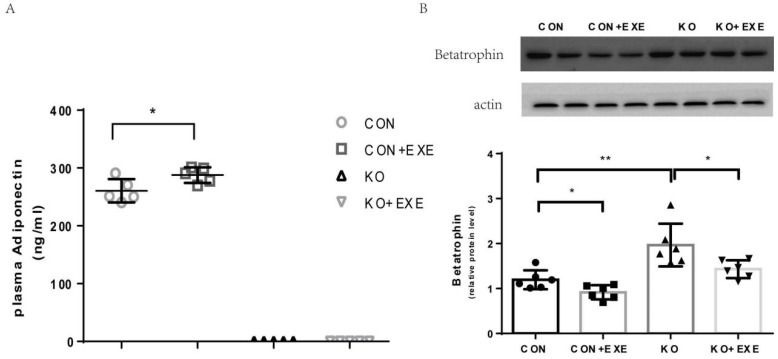
Effect of exercise on adiponectin and betatrophin expression in livers of wild type (CON) and Adipoq−/− mice. (**A**) Plasma concentrations of adiponectin and betatrophin determined using ELISA at the end of the interventions (n = 6–8 in each group). (**B**) Western blot analysis of betatrophin expression in the mouse liver. Data from at least three independent experiments are shown, and all data are expressed as the mean ± standard deviation. Student’s *t* test: * *p* < 0.05, ** *p* < 0.01.

**Figure 3 ijms-23-10630-f003:**
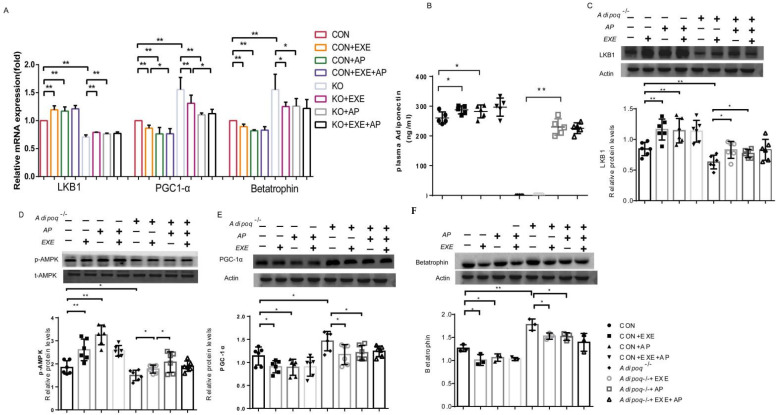
Exercise and adiponectin co-intervention reduces betatrophin expression in mouse liver. (**A**) mRNA levels of Lkb1, Ppargc1a, and Angptl8 were measured using RT-qPCR. All PCRs were performed in at least three independent experiments, in triplicate. (**B**) Plasma concentrations of adiponectin and betatrophin determined using ELISA at the end of the interventions (n = 6–8 in each group). (**C**–**F**) Western blot analysis of LKB1, p-AMPK, PGC-1α, and betatrophin expression (Wild type [CON] and Adipoq−/− mice received an i.p. injection of either Acrp30 [28 µg kg^−1^ day^−1^] or saline for 1 week. Data from at least three independent experiments are shown, and all data are expressed as the mean ± standard deviation. Student’s *t* test: * *p* < 0.05, ** *p* < 0.01.

**Figure 4 ijms-23-10630-f004:**
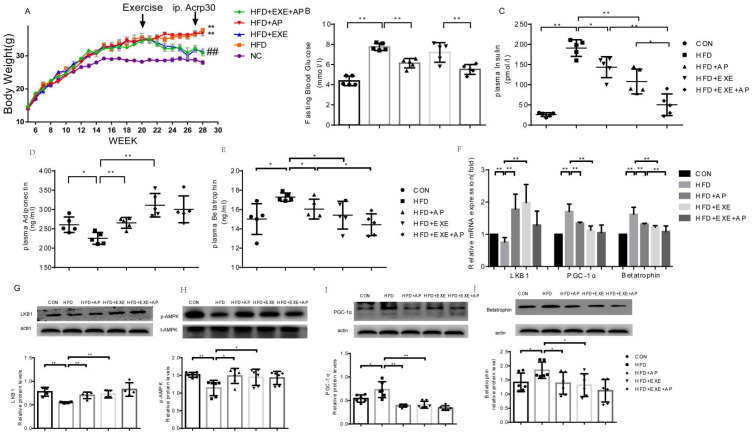
Exercise inhibits betatrophin expression and improves insulin sensitivity in obese mice. Mice fed a high-fat diet (HFD) were randomly allocated to HFD, HFD + Acrp30, HFD + EXE, or HFD + Acrp30 + EXE groups. (**A**) Body weights of HFD mice in the different intervention groups were recorded at regular intervals. (**B**) Fasting blood glucose concentration. (**C**) Serum insulin levels. (**D**,**E**) Plasma concentrations of adiponectin and betatrophin determined using ELISA at the end of the interventions (n = 6–8 in each group). (**F**) mRNA expression levels of Lkb1, Ppargc1a, and Angptl8 in the liver were measured using RT-qPCR. All PCRs were performed in triplicates in at least three independent experiments. (**G**–**J**) Western blot analysis of LKB1, p-AMPK, PGC-1α, and betatrophin expression in mouse liver. The HFD+Acrp30+EXE group received an i.p. injection of either Acrp30 (28 µg kg^−1^ day^−1^) or saline for the last week of exercise training. Data from at least three independent experiments are shown, and all data are expressed as the mean ± standard deviation. Student’s *t* test: * *p* < 0.05, ** *p* < 0.01, ^##^
*p* < 0.01.

**Figure 5 ijms-23-10630-f005:**
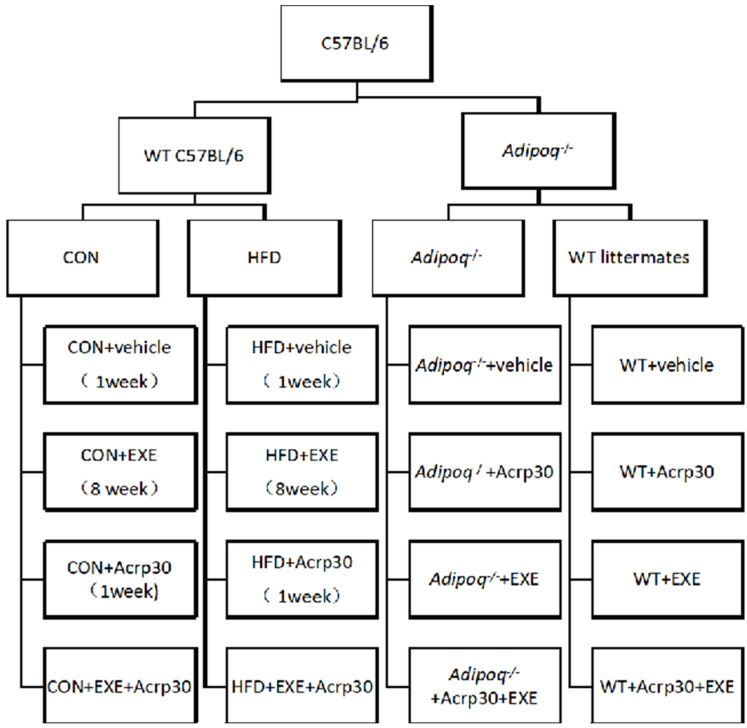
Schemata of the animal experimental set-up. CON, control; EXE, exercise; WT, wild type; Acrp30, adiponectin; HFD, high-fat diet; Adipoq, mouse adiponectin gene; Adipoq−/−, adiponectin gene knockout mice.

## Data Availability

Not applicable.

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
