# Peer review of "Adiponectin Intervention to Regulate Betatrophin Expression, Attenuate Insulin Resistance and Enhance Glucose Metabolism in Mice and Its Response to Exercise"

_ijms, 2022, doi:10.3390/ijms231810630_

Round 1

Reviewer 1 Report

The article is interesting. The authors focus on one of the most important endocrine problems, which is diabetes. They exam associations between adiponectin and betatrophin in the mice experimental set-ups.

Unfortunately, there are some limitations.

Epidemiological data on diabetes are based on publications from several years ago (2005 and 2006). Please update the literature especially on the epidemiology of diabetes.

Lots of punctuation errors and "typos" make it difficult to read the article.

To use an abbreviation, please write the full name in the first instance and follow it immediately by the abbreviated version in brackets. However, please use the abbreviations consistently following their first mention.

Author Response

Responses to Reviewer 1 Comments

Point 1:Epidemiological data on diabetes are based on publications from several years ago (2005 and 2006). Please update the literature especially on the epidemiology of diabetes.

Response 1:We agree with the reviewer’s suggestion; the introduction has been revised accordingly to include more recent publications. 

Point 2: Lots of punctuation errors and "typos" make it difficult to read the article.

Response 2: Thank you for your suggestion. We have resolved this issue to the best of our ability.We have got our manuscript proofread by a professional English editing service.

Point 3: To use an abbreviation, please write the full name in the first instance and follow it immediately by the abbreviated version in brackets. However, please use the abbreviations consistently following their first mention.

Response 3: Thank you for your suggestion. The corresponding modifications have been made.

Reviewer 2 Report

The authors showed exercise increase the adiponectin and suppress betatrophin expression in the liver.

Also, the authors showed that adiponectin or exercise attenuate impact of HFD on LKB1, AMPK and PGC1a.

It is true that adiponectin and exercise do not exert additive effect, that might suggest exercise effect is dependent on adiponectin.

However, we can see the beneficial effect of exercise such as reduction of  betatrophin and other markers in adiponectin KO mice.

These data means that adiponectin and betatrophin are independent marker during exercise.

The experiment is well done and results are sound.

But the results are not properly interpreted. So the authors should revise the title and summary to accurately reflect their findings.

In the title, the authors says “via LKB1/AMPK/PGC1”. To demonstrate this, the author should include the intervention experiment of LKB1, AMPK and PGC1a.

Figure2A, 3A; Authors should show the qPCR data. Basically, adiponectin expression in the liver tissue is very low.

Authors might have detected the plasma adiponectin. Authors should clarify the source of adiponectin in the liver.

Figure3B; As the authors mentioned in line187, this data shows exercise decrease betatrophin independently of adiponectin.

However, in the abstract (line29), the authors demonstrate that exercise reduce betatrophin via adiponectin.

The authors should correct the summary statement.

Figure4; Data are collected very well. These data also demonstrate that impact by exercise on the protein level of Lkb1, AMPK.. are regulated independently of adiponectin.

Because exercise effect can still be observed in the adiponectin KO mice.

Figure4A; Authors should correct the label PGC1? To PGC1α.

Figure5; The authors demonstrate that “adiponectin interacts synergistically with exercise” in the title. However, I can find the additional effect of adiponectin and exercise only in Fig.5C.

Most of the data in this paper suggest that there is little additional effect by adiponectin and exercise.

Figure5A; the authors should discuss the reason why fasting glucose level is high in the exercise group compared to control.

Author Response

Response to Reviewer 2 Comments

Point 1:These data means that adiponectin and betatrophin are independent marker during exercise. The experiment is well done and results are sound. But the results are not properly interpreted. So the authors should revise the title and summary to accurately reflect their findings.  In the title, the authors says “via LKB1/AMPK/PGC1”. To demonstrate this, the author should include the intervention experiment of LKB1, AMPK and PGC1a.

Response 1: We agree with the reviewer’s suggestion; accordingly, the title has been revised. Adiponectin Intervention to Regulate Betatrophin Expression, Attenuate Insulin Resistance, and Enhance Glucose Metabolism in Mice and its response to exercise

Point 2:  Figure2A, 3A; Authors should show the qPCR data. Basically, adiponectin expression in the liver tissue is very low.Authors might have detected the plasma adiponectin. Authors should clarify the source of adiponectin in the liver.

Response 2: Thank you for your question. We have indeed measured the expression of adiponectin in the liver in many experiments, but there is little literature support. Most of studies only uses immunohistochemical methods to confirm the existence of adiponectin in the liver [1-3]. In this study the level of plasma adiponectin was consistent with its expression in the liver, so we decided to replace hepatic adiponectin expression with circulating adiponectin concentration, which can also clarify the aim of this paper, that is, its relationship with exercise and betatrophin. In future studies, we will focus on the expression of adiponectin in the liver.

Point 3: Figure3B; As the authors mentioned in line187, this data shows exercise decrease betatrophin independently of adiponectin.However, in the abstract (line29), the authors demonstrate that exercise reduce betatrophin via adiponectin.The authors should correct the summary statement.

Response 3: Thank you for your suggestion. The statement in the abstract means exercise reduced Betatrophin levels via adiponectin, which modulated the LKB1/AMPK/PGC-1α signaling axis, but was not solely dependent on it for exerting its effects. The statement in line 187 attempts to explain there is a condition of an adiponectin independent. 

Point 4: Figure4; Data are collected very well. These data also demonstrate that impact by exercise on the protein level of Lkb1, AMPK.. are regulated independently of adiponectin.Because exercise effect can still be observed in the adiponectin KO mice.

Response 4: Thank you for your comment.

Point 5: Figure4A; Authors should correct the label PGC1? To PGC1α.

Response 5: Thank you for your comment. The label has been revised accordingly.

Point 6: Figure5; The authors demonstrate that “adiponectin interacts synergistically with exercise” in the title. However, I can find the additional effect of adiponectin and exercise only in Fig.5C.Most of the data in this paper suggest that there is little additional effect by adiponectin and exercise.

Response 6: We agree with the reviewer’s suggestion; the title has been revised accordingly. Adiponectin Intervention to Regulate Betatrophin Expression, Attenuate Insulin Resistance and Enhance Glucose Metabolism in Mice and its response to exercise.

Point 7: Figure5A; the authors should discuss the reason why fasting glucose level is high in the exercise group compared to control.

Response 7: The HFD+EXE group in Fig. 5 is the exercise group after the high-fat diet model. Exercise intervention is performed based on the increase in blood glucose. Compared with the high-fat diet control group, in the exercise group, the fasting blood glucose improved, but it did not return to the level observed in the normal diet group.

  1. Ma, H.; Gomez, V.; Lu, L.; Yang, X.; Wu, X.; Xiao, S.Y. Expression of adiponectin and its receptors in livers of morbidly obese patients with non-alcoholic fatty liver disease. J Gastroenterol Hepatol. 2009, 24, 233–237. DOI:10.1111/j.1440-1746.2008.05548.x.
  2. Cai, J.; Hu, Q.; Lin, H.; Zhao, J.; Jiao, H.; Wang, X.. Adiponectin/adiponectin receptors mRNA expression profiles in chickens and their response to feed restriction. Poult Sci. 2021, 100, 101480. DOI:10.1016/j.psj.2021.101480.
  3. Uribe, M.; Zamora-Valdés, D.; Moreno-Portillo, M.; Bermejo-Martínez, L.; Pichardo-Bahena, R.; Baptista-González, H.A.; Ponciano-Rodríguez, G.; Uribe, M.H.; Medina-Santillán, R.; Méndez-Sánchez, N. Hepatic expression of ghrelin and adiponectin and their receptors in patients with nonalcoholic fatty liver disease. Ann Hepatol. 2008, 7, 67–71. DOI:10.1016/S1665-2681(19)31890-3.

Round 2

Reviewer 2 Report

Well corrected.